# Carbonized Dehydroascorbic Acid: Aim for Targeted Repair of Graphene Defects and Bridge Connection of Graphene Sheets with Small Size

**DOI:** 10.3390/nano10030531

**Published:** 2020-03-16

**Authors:** Jing Li, Jinfeng Lai, Jialiang Liu, Rubai Lei, Yuxun Chen

**Affiliations:** 1School of Chemistry and Chemical Engineering, South China University of Technology, Guangzhou 510641, China; laijinfeng666@126.com (J.L.); leirubai@foxmail.com (R.L.); 2South China University of Technology-Zhuhai Institute of Modern Industrial Innovation, Zhuhai 519000, China; 3School of Mechanical and Automotive Engineering, South China University of Technology, Guangzhou 510641, China

**Keywords:** targeted repair, oxidized graphene, carbonized dehydroascorbic acid, thermal reduction, thermal conductivity

## Abstract

The thermal dissipation issue of electronics devices becomes increasingly prominent as they evolve to smaller sizes and more complicated structures. Therefore, the development of materials with excellent heat conduction properties and light weight turns out to be an urgent demand to solve the heat transfer problem of electronics devices with high performance. For this purpose, we put forward an innovative strategy that carbonized dehydroascorbic acid (CDA) be applied to graphene layers for the targeted repair of defects among them and bridge connection of the layers to produce graphene heat conduction materials with excellent properties. Firstly, hydrogen bonds formed from dehydroascorbic acid (DHAA, products of the oxidation of vitamin C) and each of ketone, carboxyl, and oxhydryl groups on graphene layers were absorbed at targeted locations where oxidation graphene produces defects, then targeted repair was conducted for those defects to be filled and for the graphene layers of a small size to grow into large sheet materials with improved continuity by CDA generated in thermally pressing reduction reaction at 800 °C. In our investigation, the planar thermal conductivity of rGO/VC membrane reached 1031.9 ± 10.2 Wm^−1^K^−1^, while the added mass content of vitamin C (VC) was 15%. Being a reference, the planar thermal conductivity of primitive graphene membrane was only 610.7 ± 11.7 Wm^−1^K^−1^.

## 1. Introduction

As microelectronics technology develops rapidly, various electronic devices have evolved with light weight, small size, and high performance leading to higher and higher integration levels in integrated circuits. This gives rise to an increasingly prominent problem of heat dissipation. Predicted with moore’s law, the density of transistors in a heavily packed integrated circuit may double every 18 months [1,2]. If the heat accumulated in an electronic device cannot be dissipated during operation after long periods, its temperature can rise to such a level that its operational stability and life span may be affected [3]. One solution for this problem is to develop materials with higher thermal conductivity through which unwanted heat in electronic devices can dissipate, so that the temperature of the integrated circuit decreases. Traditional thermal materials, such as silver, copper, aluminum, etc., have the disadvantages of high density, low thermal conductivity, and high production cost, and hence cannot cope with the increase in the heat dissipation demand of integrated circuits which is very significant for the development of materials with excellent thermal conductivity, flexibility, and light weight.

Graphene (Gr), a two-dimensional nanomaterial of *sp*^2^-hybridized carbon atoms arranged in a hexagonal honeycomb lattice structure [4], has excellent flexibility [5], and it is a carbon material with the highest thermal conductivity ever known. For instance, the thermal conductivity of monolayer graphene without defects can be as high as 5300 Wm^−1^K^−1^ [6,7], far beyond that of copper, silver, diamond, and carbon nanotubes, etc. Therefore, after being assembled into two-dimensional macroscopic sheets, it possesses very wide prospects for applications and can be considered as a replacement for normal thermal conduction materials [8]. However, the direct application of single or few-layer graphene cannot meet the practical heat transfer requirements of electronic devices owing to the non-freestanding structures, size limitation, and limited heat flux carrying capacity. Therefore, graphene films composed of graphene sheets are promising candidates as thermal management materials [9,10]. Graphene oxide (GO), as a precursor compound of graphene, has good water and organic dispersibility [11,12,13]. It can be prepared into macroscopical GO films with low thermal conductivity using simple methods such as solvent evaporation. The GO films are then subjected to thermal reduction to obtain graphene films (GFs, also known as reduced graphene oxide (rGO) films) with high thermal conductivity. Shen et al. [14] fabricated GO films by solvent evaporation, and then thermally reduced the films at 2000 °C to obtain graphene films with an in-plane thermal conductivity of 1100 Wm^−1^K^−1^. Huang et al. [15] prepared large-sized, free-standing, and flexible rGO films with a high thermal conductivity of 826 Wm^−1^K^−1^ by high-temperature annealing (2800 °C) of GO films. The thermal conductivity of these graphene films was lower than the theoretical thermal conductivity of graphene. This is because the graphene sheets constituting these graphene films are imperfect and contain many defects such as mono-vacancies, pentagonal/heptagonal rings, and oxygen-containing functional groups [16,17]. These defects can reduce the thermal conductivity of graphene significantly (~83%) even at an extremely low defect concentration (~0.1%) because of an increase in defect-induced phonon scattering [16,17,18,19]. Li et al. [20] fabricated graphitized graphene/polyimide hybrid films with high thermal conductivity using polyimide (PI) to repair graphene sheet defects at 2800 °C The resulting films exhibited enhanced thermal conductivity (from ~650 Wm^−1^K^−1^ to ~800 Wm^−1^K^−1^). Wang et al. [21] observed that polyacrylonitrile (PAN) can be used to repair graphite oxide (GtO) by graphitization at 2850 °C, and the maximum in-plane thermal conductivity of graphitized GtO/PAN films was enhanced by ~18% compared to the graphitized GtO films. However, the temperature required for graphitization is extremely high which results in large energy consumption, high cost, and complex equipment. Therefore, it will be beneficial to prepare graphene films with a higher thermal conductivity by repairing the defects in graphene sheets and bridge connection of graphene sheets of a small size at a moderate temperature (<1000 °C).

Here, we propose a new strategy for the targeted repair of the internal defects in graphene layers and bridge connection of graphene sheets of small size with carbonized dehydroascorbic acid (CDA) produced during decomposition of heated dehydroascorbic acid (DHAA). During the preparation of GO/VC membrane, Vitamin C (VC) reduces partial oxygen-containing groups of graphene oxide and produces DHAA [22]. The DHAA distribute evenly among graphene oxide films (GOFs) layers and conduct targeted repair for the defects among graphene layers generated during the thermally pressing reduction of ketone, carboxyl, and hydroxyl at 800 °C. Besides, CDA generated by decomposition of heated DHAA can further provide bridge connection to graphene sheets of small size for larger sheet size, better connectivity, and less phonon transmission among layers so as to reduce the interlayer thermal resistance. When the mass content of added vitamin C was 15%, the planar thermal conductivity of rGO/VC film can reach the maximum value of 1031.9 ± 10.2 Wm^−1^K^−1^, which was increased by 69.0% compared with the original GFs. Therefore, we provide a promising method for preparing graphene paper with excellent heat transfer properties which is expected to be widely used in the field of thermal management and heat dissipation of electronic devices.

## 2. Materials and Methods

### 2.1. Collection of Raw Materials

In our current study, natural graphite powders (325 mesh) were purchased from Tengshengda Tansu Jixie Co. Ltd., Qingdao, China. Vitamin C was purchased from Shanghai Bao Biotechnology Co. Ltd. (Shanghai, China). Potassium permanganate (99%), sulfuric acid (98%), concentrated hydrochloric acid (36%), and hydrogen peroxide (30%) were purchased from Guangzhou Chemical Reagents Factory (Guangzhou, China).

### 2.2. Preparation of GO/VC and rGO/VC-x% Films

The GO was prepared using the “two-step” modified Hummers method (Appendix A) [23], and the GO suspension (8.0 mg mL^−1^) was treated in ultrasonic cleaner (Shanghai Kedao SK7200H, Shanghai, China) at 53 kHz and 350 W for 60 min. The preparation of rGO/VC-15% films mainly took four steps. Firstly, GO/VC-15% solution (250 mL) was prepared by adding 35.29 mg of vitamin C to GO suspension at room temperature and then placed motionlessly for three days for a preliminary chemical reduction treatment to obtain the intermediate products GO/DHAA-15% solution. Vitamin C was added to reduce GO for the restoration of some of the damaged *sp*^2^ structure among carbon atoms (Appendix A), the production of DHAA, and the initial reduction of GO. Secondly, the GO/DHAA-15% solution was heated to 80 °C and stirred for half an hour to ensure that DHAA was fully mixed with GO sheets. Thirdly, GO/DHAA-15% solution was directly transferred into an acrylic plexiglass groove (22 × 30 × 1 cm) to have the solvent evaporated at 50 °C for 8 h in a vacuum drying oven. The important thing to note here is that the films thus obtained was designated as GO/VC-15% film, with a thickness of 0.018 mm. The name of the GO/VC samples represented GO membranes prepared by the addition of vitamin C. Finally, the GO/VC-15% film was loaded into a self-made thermal reduction mold, and then both were placed in a high-temperature tube furnace (SLG1200-100, Shanghai Shengli Instruments Co. Ltd., Shanghai, China). The sample was annealed by TPR (temperature-programmed reduction) (Appendix A) in an argon (purity, 99.999%) flow of 50 mL min^−1^. Subsequently, the sample was naturally cooled down to ambient temperature in the argon flow (about 3 h). In the same manner, we fabricated the GO/VC-*x*% and rGO/VC-*x*% samples, where *x* represents the mass content percentage of vitamin C to GO.

### 2.3. Characterization

The morphology of the film samples was characterized using scanning electron microscopy (SEM) with a Hitachi UHR FE-SEM SU8200 system (SU8200, Hitachi, Tokyo, Japan) operated at 10.0 kV. The high-resolution transmission electron microscopy (JEOL-JEM 2100F, JEOL, Tokyo, Japan) was used to reveal the microstructure of the samples. The chemical bonds and element contents of the samples were analyzed by X-ray photon energy spectrum (XPS) with a Sigma probe and monochromatic X-ray source. The samples were pressed with KBr powders to form pellets for testing. Fourier transform infrared spectrometer (Tensor 27, Bruker, Bremen, Germany) was used to analyze the functional group change before and after thermal reduction. At the heating rate of 10 °C/min, the samples were subjected to thermogravimetric test (NETZSCH TG209F3, Netzsch, Selb, Germany) in nitrogen flow, and the thermal stability and decomposition process of the samples were to be analyzed. X-ray diffraction (D8 Advance, AXS, Karlsruhe, Germany) with Cu-Ka radiation (k = 0.15406 nm) was performed over the range of 5°–50°. Laser confocal Raman microscopy was performed using a Horiba Jobin-Yvon LabRAM ARAMIS system (LabRAM Aramis, H.J.Y, Paris, France) with He–Ne laser excited at 532 nm. The electrical conductivity was measured at room temperature by the standard four-point probe method (Agilent 34420A, Loveland, CO, USA). The samples were prepared as strips with a width of 3 mm and length of approximately 5 cm.

Infrared camera (FLIR MSX TECHNOLOGY, FLIR, Boston, MA, USA) was utilized to identify the heat dissipation efficiency of samples and record the temperature profiles. We set-up a thermal test platform for heating the center of the sample with a probe and recording the temperature distribution on the surface of the sample with an infrared camera.

### 2.4. Measurement of Thermal Conductivity

The planar thermal diffusivity of all samples was determined by a laser flash (LFA 447, Netzsch, Germany) at room temperature (30 °C). First, the film was cut into a circular sample with a diameter of 25.4 mm, then the measurement of the sample was conducted with a laser flash meter, during which the sample was heated with light pulses and the temperature rise at four different locations was measured with an infrared detector. The following relationship between temperature and time was analyzed to determine:*α* = 0.1388 × *d*^2^ × *t*_1/2_^−1^(1)
where *α* is the thermal diffusivity, *d* is the thickness of the tested sample, and *t*_1/2_ is half of the diffusion time. Thermal conductivity (*λ*) was calculated by the standard ASTME 1461-92 method and was obtained using Equation (2).
*λ* = *C_p_* × *ρ* × *α*(2)
where *λ* is the thermal conductivity of the sample (Wm^−1^K^−1^). *C_p_* is the specific heat capacity of the sample at constant pressure (J/kg K). Netzsch DSC 204 F1 was obtained from the test at the scanning temperature of −20 °C–120 °C; *ρ* is the density of the sample (g/cm^3^) which can be calculated by the mass, diameter, and thickness of the thin-film sample; and *α* represents the thermal diffusion coefficient of the sample (m^2^/s). The mass of the sample was obtained using an electronic balance of 1/10,000. To prepare the sample, the thin film was cut into a round sample with a diameter of 25.4 mm. An electronic digital display micrometer caliper was used to measure the thickness of the sample at 5 different positions and calculate the average value.

## 3. Results and Discussion

### 3.1. Morphology and Structural Characterizations

Preparation of the rGO/VC films with a two-step reduction method is shown in Figure 1a. Vitamin C was added to reduce GO for the restoration of some of the damaged *sp*^2^ structure between carbon atoms (Appendix A), the production of DHAA, and the initial reduction of GO. Although compared with the traditional stronger reducing reagent, Vitamin C was less effective than reducing agents such as hydroiodic acid and hydrazine hydrate; it is an environmentally friendly reducing reagent, and its composition of the elements was consistent with the graphene oxide which is helpful to avoid generation of phonon scattering caused by impurity atoms (such as N, I, S) [24]. Evenly distributing between GO layers, the DHAA molecules formed hydrogen bond with ketone, carboxyl, and hydroxyl groups on targeted locations where the defects existed so that they could be absorbed on GO layers and provided precise locations for CDA to repair those defects. The GO/VC film with a size of 22 × 30 cm was further prepared by evaporating the solvent (Appendix A), and then small pieces of the GO/VC film were cut and thermally pressed to obtain the rGO/VC film. Features of GO suspension are shown in Figure 1b after it was reduced by vitamin C. There was no change in the color of the GO solution (0.5 mg/mL, solvent was water) after standing still for three days at room temperature while the GO/VC solution (the mass content percentage of vitamin C to GO was 15%) obviously turns dark to produce the GO/DHAA solution. Analysis of the color change of the GO/VC solution further confirmed that the chemical reduction reaction occurred between vitamin C and GO. Optical photos of GO/VC and rGO/VC films are shown in the Figure 1. The GO/VC film (Figure 1c,d) was black in color with a very smooth surface. In addition, the GO/VC film could be arbitrarily bent, wound, and repeatedly folded without cracking. The interface was shaped by the size of the evaporation mold during the self-assembly process. The rGO/VC films (Figure 1e,f) can be obtained after mechanically pressing and thermal annealing at 800 °C. Their surfaces are of metallic luster and no cracks or fractures were found when folded with tweezers, showing their perfect flexibility.

Under SEM, the microscopic cross-sections of GO and GO/VC-15% films are observed in Figure 2a,b. It can be seen that there were layered structures arranged neatly with high compactness and basically no air pockets between the layers. The images (Figure 2c) show that the graphene nanosheet layers were stacked unevenly with obvious air pockets. The SEM surface of the rGO film (Figure 2e) was rough and uneven with obvious deep grooves, cracks, and folds. This is because the thermal decomposition of GO mainly occurs at 180–220 °C, rapidly producing a large amount of CO_2_, CO, and H_2_O gases between graphene layers. The failure of these gases to escape in time causes greater gas pressure build-up between the graphene layers, resulting in greater damage to the rGO structure. The presence of folds and chaotic lamination stack can significantly enhance phonon scattering in the planar direction, reducing the planar thermal conductivity of rGO. However, the rGO/VC-15% film with vitamin C (Figure 2d,f) had a relatively neat laminate structure and high continuity, and the cracks were repaired to a great extent in rGO/VC-15% film and displayed an integrated surface with some ripples. The rGO/VC-15% film had a relatively wide reduction temperature range and gas was produced at a slower pace. In addition, DHAA was first decomposed thermally to produce CO_2_, CO, H_2_O, and other gases which evenly spread out the graphene sheet to form a gas channel to help gas emission so as to obtain GFs with orderly arrangement and a smooth surface. Such well-arranged laminate structure with fewer defects can minimize phonon scattering and improve heat transfer performance [25]. The thickness of the two samples was 14 μm and 15 μm, respectively. The TEM of rGO and rGO/VC-15% is shown in Figure 2g,h. The surface of rGO/VC-15% film was smoother and the connection between layers was more compact. The SAED pattern of the rGO/VC-15% was more pronounced, exhibiting a better six-fold lattice structure [26]. This shows that the rGO/VC-15% film had a more complete graphite crystal domain structure. This was because CDA formed from DHAA-repaired graphene defects effectively during thermal reduction and thus obtained rGO/VC-15% film with a better π-conjugated structure and lower lattice chaos.

Further, the chemical bonding and element content were characterized using XPS. As shown in Figure 3a,b, the C/O ratios of rGO and rGO/VC-15% films were l6.83 and 23.66, respectively. This indicates that the latter film had a higher carbon content and lower oxygen content. This is attributed to the decomposition of DHAA during thermal reduction to produce CDA which could be used as an external carbon source to compensate for lost carbon atoms on the graphene sheets due to the decomposition of oxygen-containing functional groups. This could repair the structural defects and improve the structural integrity of graphene sheets, thereby reducing the degree of damage to the graphene structure by temperature. The C 1s spectra of the rGO and rGO/VC-15% films can be deconvoluted into five peaks arising from *sp*^2^ C (~284.6 eV), *sp*^3^ C (~285.2 eV), –C–O (~286.8 eV), –C=O (~287.4 eV), and –COO (~288.8 eV) [25]. Shifts of whole C 1s spectra were influenced by electron-withdrawing functional groups such as C=O-containing functional groups. Full-width at half maximums of the main peak of C 1s spectra were influenced by mainly electron-withdrawing functional groups in addition to defects such as vacancy, pentagons, and heptagons [27]. In contrast to rGO, the peak associated with *sp*^3^-hybridized C and –C–O chemical composition clearly reduced in rGO/VC-15%, confirming that vitamin C had a chemical reduction effect of epoxy groups and hydroxyl groups on the GO and eliminated part of the oxygen-containing groups before thermal reduction which partially restored the damaged *sp*^2^ structure among carbon atoms. In addition, it also showed that a large number of defects were effectively repaired during thermal annealing due to the presence of DHAA. The heat conduction of graphite materials is mainly controlled by the efficiency of phonon transmission along *sp*^2^ bonded carbon lattice [28,29,30]. The higher content of *sp*^2^ structure of graphene was useful to reduce phonon scattering, thus improving the thermal conductivity of rGO.

Thermogravimetric analysis of GO, GO/VC-15% and VC is shown in Figure 3c. The thermal decomposition of GO/VC-15% mainly takes place at 100–300 °C, so a prolonged heating process can effectively reduce the pressure build up generated by thermal decomposition of oxygen-containing functional groups and evenly separate the stacked graphene sheets to form a structure with orderly layers. The oxygen-containing groups and residual solvents increase the graphene sheet spacing [31], and the lower thermal reduction temperature and reduction efficiency cannot effectively remove the oxygen-containing functional groups between the graphene sheets, thus increasing the thickness of the graphene film. At 800 °C, the mass decomposition of GO/VC-15% was stable, indicating that the addition of vitamin C accelerates thermal decomposition of oxygen-containing functional groups on GO, thereby lowering the thermal decomposition temperature. Infrared spectra are shown in Figure 3d to study the changes in functional groups of GO and GO/VC-15% before and after thermal reduction. Compounds formed by carboxyl, phenol, and intercalated H_2_O shows a wide hydroxyl stretching-compressing vibration mode at 3400 cm^−1^ [32]. At 1732 cm^−1^, 1606 cm^−1^, 1390 cm^−1^, 1041 cm^−1^, the characteristic peaks are respectively shown as the stretching–compressing vibration peaks of C=O, C=C, C–OH, and C–O–C groups [33,34]. The oxygen-containing functional groups of vitamin C (C_6_H_8_O_6_) used in the experiment included hydroxyl, epoxy group, and ketone group, and there were no heteroatoms, so the GO/VC-15% compound film after the addition of vitamin C had no other oxygen-containing functional groups compared with the GO film. C=O, C=C, C–OH, C–O–C, and other oxygen-containing groups at 1732 cm^−1^, 1606 cm^−1^, 1390 cm^−1^, 1041 cm^−1^ of rGO and rGO/VC-15% obtained by thermal reduction almost disappeared, and only an obvious characteristic peak of C=C appeared at 1610 cm^−1^ with weaker intensity.

The XRD patterns of Gr, GO, GO/VC-15%, rGO, and rGO/VC-15% are shown in Figure 3e. The diffraction peak of graphene (Gr) appeared at 26.69°, corresponding to an interlayer spacing of 0.335 nm. The diffraction peak of GO film appeared at 10.26°, corresponding to an interlayer spacing of 0.861 nm due to the attachment of oxygen-containing functional groups and residual solvents in the GO layers [35]. The diffraction peak of GO/VC-15% film appeared at 11.25°, corresponding to an interlayer spacing of 0.785 nm, demonstrating that vitamin C partially reduced oxygen groups of GO partially. After thermal reduction, rGO and rGO/VC-15% had higher crystallinity, and the diffraction peaks becamme 25.67° and 26.46°, corresponding to an interlayer spacing of 0.347 and 0.337 nm. After thermal reduction, the diffraction peaks of rGO and rGO/VC-15% were relatively short and wide [36,37], and the diffraction peaks became 25.67° and 26.46°, corresponding to an interlayer spacing of 0.347 and 0.337 nm. This is because GO is only carbonized at a mild thermal reduction temperature of 800 °C, and the lower thermal reduction temperature was not sufficient to achieve graphitization. Therefore, the crystallinity of rGO was not high, and the diffraction peak was relatively short and wide. Combined with XPS and FTIR spectroscopy analysis, most oxygen-containing groups of rGO and rGO/VC were eliminated after thermal reduction at 800 °C, and with a small number of oxygen-containing groups such as carbonyl group and quinone remaining. These oxygen-containing groups could be completely removed only at a higher temperature [38]. These remaining groups act as a hindrance to the stacking of graphene sheets, so rGO and rGO/VC-15% were distributed with slightly more spacing than natural Gr (0.335 nm). However, CDA produced by the thermal decomposition of DHAA was favorable for the aggregation of graphene sheets and the production of more compact graphene layer, so rGO/VC-15% was less separated than rGO sheets.

With Raman spectroscopy, the structural integration of the original rGO, rGO/VC-*x*%, and GO/VC-15% was further investigated. As shown in Figure 3f, two characteristic peaks can be observed: the D peak at 1346 cm^−1^ and G peak at 1592 cm^−1^. The strength ratio of the D peak to G peak is considered to be related to the density of graphene defects [39]. The generation of the D peak is attributed to defects in the carbon structure, while the G peak is believed to be caused by the first-order scattering of *sp*^2^ hybrid carbon atoms in the E2g mode [40,41]. If the intensity of the G peak is stronger than that of the D peak, it indicates a higher degree of disorder in the graphene layer. The average domain size of *sp*^2^ hybrid carbon atoms can be calculated by Cançado’s equation [42]. From the Raman spectrum in Figure 3g, the chemically reduced graphene vitamin C also had fundamental vibrations at 1350 cm^−1^ (D band) and 1590 cm^−1^ (G band). The fitting of the observed Raman spectra using the Raman spectrometer software gives the D/G intensity ratios. The ratio *I_D_*/*I_G_* of original GO/VC-15% film was 1.74, corresponding to a crystal domain size of 11.05 nm. After thermal reduction, the *I_D_*/*I_G_* ratio of rGO and rGO/VC films (Figure 3h) decreased significantly, and the crystal domain size La increased obviously. This change suggests that more *sp*^2^ domains were formed during the thermal reduction of graphite oxide. When the mass content of vitamin C was 15%, the *I_D_*/*I_G_* of rGO/VC-15% reached the minimum value of 0.85 (*I_D_*/*I_G_* of rGO was 1.2), corresponding to a crystal domain size reaching the maximum value of 22.62 nm (crystal domain size of rGO was 16.00 nm). The decrease in *I_D_*/*I_G_* ratio was due to the reduction of the quantity of disordered carbon atoms and defects generated in the thermal decomposition process [41]. Fewer defects can effectively reduce phonon scattering to enhance the thermal conductivity of graphene films. During the carbonization process, oxygen-containing groups on the graphene nanosheet layers are heated and produce CO_2_ (carbon atoms come from GO), so as to produce more defects [43,44] such as single-vacancy defects, double-vacancy defects, stone-wales defects, and linear defects. Carbonized dehydroascorbic acid can effectively repair defects of graphene sheets and connect sheets with small size to form large size graphene sheets to enhance interlayer integration and continuity. Therefore, the ratio of rGO/VC films was smaller than rGO films. The increase in the size of the domain was conducive to the transmission of low-frequency ballistic phonons in the domain and the better transmittance on the grain boundary [45], resulting in the higher planar thermal conductivity of rGO/VC-15%. However, the addition of too much vitamin C tends to produce the aggregation of activated carbon substances, thus resulting in the increase of *I_D_*/*I_G_* ratio which is not conducive to the enhancement of interlayer continuity and integration. Therefore, it is necessary to add vitamin C of an optimal amount to improve the thermal conductivity of the film.

### 3.2. Thermal Conductivity and Infrared Surface Thermography of Modified Graphene Films

The planar thermal conductivity of rGO films and rGO/VC films with various amounts of vitamin C are measured with laser emission (Netzsch LFA 447). The density and thickness of each sample are listed in Appendix A. The planar thermal conductivity of rGO film and rGO/VC films with various amounts of vitamin C added was computed with the formula of thermal conductivity as shown in the Figure 4a. At room temperature, the thermal conductivity of the original GFs was only 610.7 ± 11.7 Wm^−1^K^−1^, while the planar thermal conductivity of rGO/VC films with various vitamin C contents increased first and then decreased as the Vitamin C content increased. Because DHAA (products of the oxidation of vitamin C) was first decomposed thermally to produce CO_2_, CO, H_2_O, and other gases, it evenly spread out the graphene sheet to form a gas channel to help gas emission, thus reducing the chaotic stacking of the graphene sheets. However, the excess vitamin C will produce over atmospheric pressure, resulting in the formation of air holes between graphene sheets, which destroy the ordering of graphene sheets. On the other hand, DHAA thermally decomposed CDA not only repaired the vacancy defects of graphene but also produced amorphous carbon. The presence of an appropriate amount of amorphous carbon was conducive to bridging graphene sheets, enhancing interlaminar continuity and structural integrity, while excess amorphous carbon will hinder the repair of defects by CDA and bridging graphene sheets. When the mass content of added vitamin C was 15%, the planar thermal conductivity of rGO/VC film reached the maximum value of 1031.9 ± 10.2 Wm^−1^K^−1^, which was increased by 69.0% compared with that of the original GFs. The heat transfer simulation (Appendix A) also showed that it had good heat dissipation effect. This value was also much higher than that of pure copper (about 400 Wm^−1^K^−1^) [46] or directional carbon nanotubes (776 Wm^−1^K^−1^) [47]. When the mass content of vitamin C was 5%, 10%, 20%, and 25%, respectively, the planar thermal conductivity of the film was 805.2 ± 8.9 Wm^−1^K^−1^, 901.3 ± 12.3 Wm^−1^K^−1^, 928.6 ± 26.3 Wm^−1^K^−1^, and 893.7 ± 8.8 Wm^−1^K^−1^, respectively. In addition, the electrical conductivity of GO films was only 0.8 S/cm, while the rGO films increased to 453 S/cm. The electrical conductivity of rGO/VC films with various vitamin C contents rose first and then declined as the vitamin C content increased (Figure 4b). When the mass content of added vitamin C was 15%, the electrical conductivity of rGO/VC film reached the maximum value of 1396 S/cm. These results are in accordance with the improvements observed in the thermal properties. It is known that defects and incomplete reduction hinder the electrical performance of rGO in terms of both reduced electrical conductivity and charged carrier mobility and the thermal treatment favors the increase of *sp*^2^ domains [48]. As shown in the Figure 4, the thermal images of GO/VC-15%, rGO, and rGO/VC-15% films shows that GO/VC-15% films only treated with chemical reduction (Figure 4c) had poor average temperature performance, heat at the central heating point could not be transferred to the edge of the film quickly, and the planar thermal conductivity was low. The rGO film treated only by thermal reduction (Figure 4d) had a temperature of 53.1 °C at the center heating point and 40 °C at the edge, with its thermal conductivity and homogenization greatly improved. After chemical reduction and thermal reduction, the temperature of rGO/VC-15% film at the center was close to that at the edge (41.1 °C) (Figure 4e), and its thermal conductivity and homogenization was the best. The results are consistent with the results obtained by the test for the planar thermal conductivity of rGO/VC films with various vitamin C content added.

### 3.3. The Mechanism for the Function of Carbonized Dehydroascorbic acid for Targeted Repair of Graphene Defects and Bridge Connection of Graphene Sheets with Small Size

The mechanism for the function of CDA for targeted repair of graphene defects and bridge connection of graphene sheets with small size is shown in Figure 5. The preparation process of the original rGO membrane is shown in route 1. It can be seen that after thermal reduction at 800 °C, there were many defects in the graphene structure as well as loose interlayer connections and uneven stacking. The preparation of rGO/VC films with thermal reduction at 800 °C is shown in route 2. It can be seen that DHAA distributing evenly among GO films layers formed hydrogen bonds with ketone, carboxyl, and hydroxyl [22]. Uniformly mixing among GO layers, DHAA formed hydrogen bonds with keto, carboxyl, and hydroxyl groups, so that it was absorbed at targeted locations where defects among GO layer appeared. CDA, which is generated during the thermal decomposition of DHAA, conducts targeted repair of defects among GO layers and effectively recovers the damaged *sp*^2^ hybrid structure for better integration of GO films. In addition, CDA generated by its thermal decomposition fills the gap between the small graphene sheets and act as a bridge between the sheets. Bridging adjacent graphene sheets with CDA enhances overall continuity of the graphene membrane and provides a channel for phonon transport, thereby reducing interlaminar thermal resistance and resulting in higher thermal conductivity of the rGO/VC films.

## 4. Conclusions

In this work, with the reference to Gao et al.’s [49] proposal to add vitamin C to GO for a preliminary chemical reduction to remove part of oxygen-containing groups, we developed a new method combined with a thermal reduction process to repair graphene layer defects. Therefore, it is a pioneering research work for the preparation of highly conductive graphene films under mild process conditions of chemical reduction and lower thermal annealing temperatures. Compared with other methods to prepare GFs of high thermal conductivity, lower thermal annealing temperatures were accomplished for energy savings, environmental protection, and also reduction of the production cost of GFs of similarly high thermal conductivity. Vitamin C and GO undergo a REDOX reaction that restores part of the damaged *sp*^2^ structure among carbon atoms, produces DHAA, and has GO primarily reduced. Besides, dehydrogenated ascorbic acid molecules, evenly distributing between GO layers, form hydrogen bonds with ketone, carboxyl, and hydroxyl groups on the layers for targeted defect locating and provided a precise location for CDA to repair defects during thermal reduction. At the same time, CDA generated by thermal decomposition of DHAA fills the gaps among the small graphene layers and acts as a bridge to enhance the overall continuity of the graphene film which provided channels for phonon transportation to improve the heat transfer performance of GFs. Based on the measurement, it was discovered that while the mass content of added vitamin C was 15%, the planar thermal conductivity of rGO/VC films reached the maximum value of 1031.9 ± 10.2 Wm^−1^K^−1^ which increased by 69.0% compared with the original GFs. In conclusion, the preparation of high thermal conductivity GFs by this two-step reduction method provides a promising way for the preparation of graphene paper and promotes the potential for application of graphene paper in thermal management.

## Figures and Tables

**Figure 1 nanomaterials-10-00531-f001:**
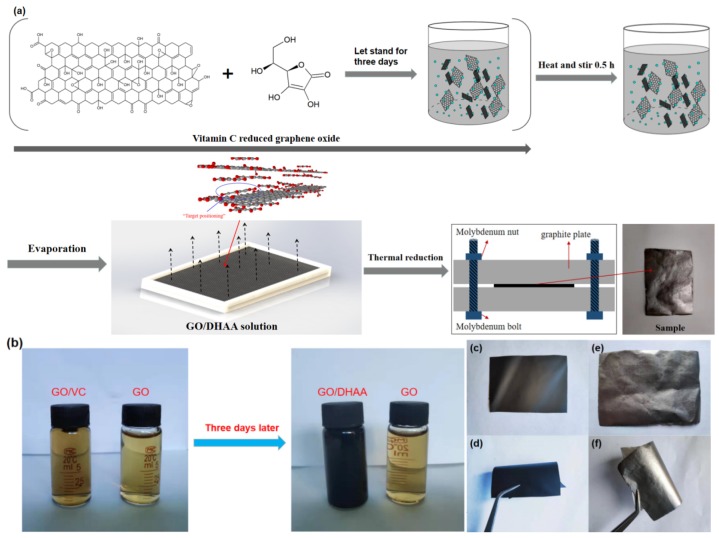
(**a**) Flow chart of the rGO/VC (reduced graphene oxide was added with vitamin C) film preparation by solvent evaporation; (**b**) before and after chemical reduction of vitamin C for GO; (**c**,**d**) optical photo of GO/VC; (**e**,**f**) rGO/VC-15% optical photo.

**Figure 2 nanomaterials-10-00531-f002:**
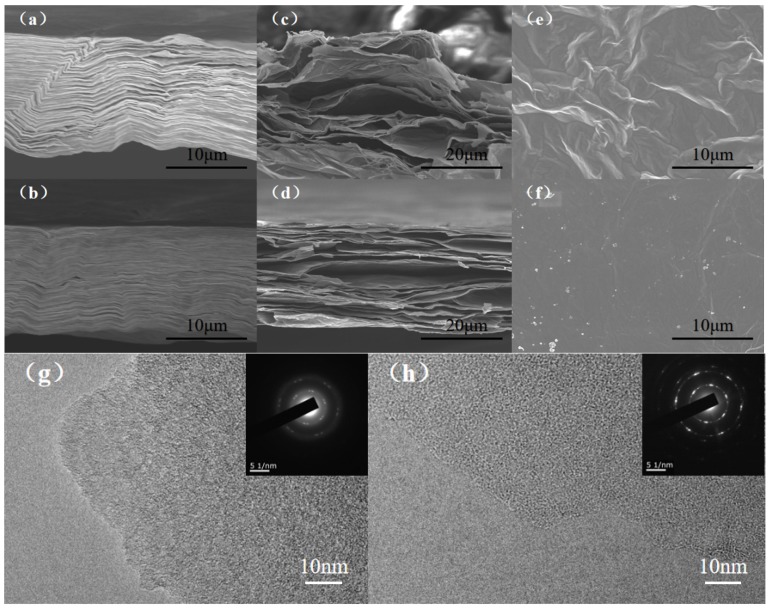
(**a**,**b**) Cross-sections of GO and GO/VC-15%; (**c**,**e**) cross-sections and surfaces of rGO; (**d**,**f**) rGO/VC-15% cross-sections and surfaces; (**g**,**h**) TEM characterization of rGO and rGO/VC-15%.

**Figure 3 nanomaterials-10-00531-f003:**
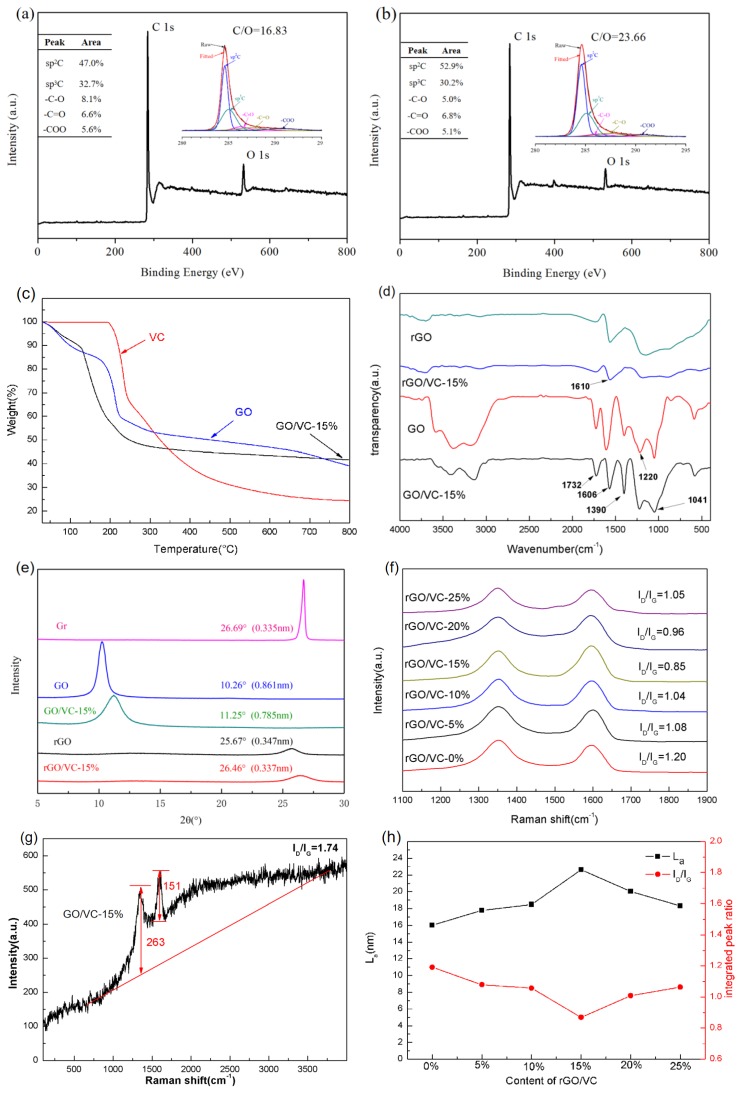
(**a**,**b**) XPS analysis charts of rGO and rGO/VC-15%; (**c**) the thermal gravimetric analysis of GO, GO/VC-15%, and VC; (**d**) the infrared spectra of GO, GO/VC-15%, rGO and rGO/VC-15%; (**e**) XRD of Gr, GO, GO/VC-15%, rGO and rGO/VC-15%; (**f**,**g**) Raman spectrum of rGO, rGO/VC-15% and GO/VC-15%; (**h**) I_D_/I_G_ of rGO, rGO/VC-15% and the size of the crystal field.

**Figure 4 nanomaterials-10-00531-f004:**
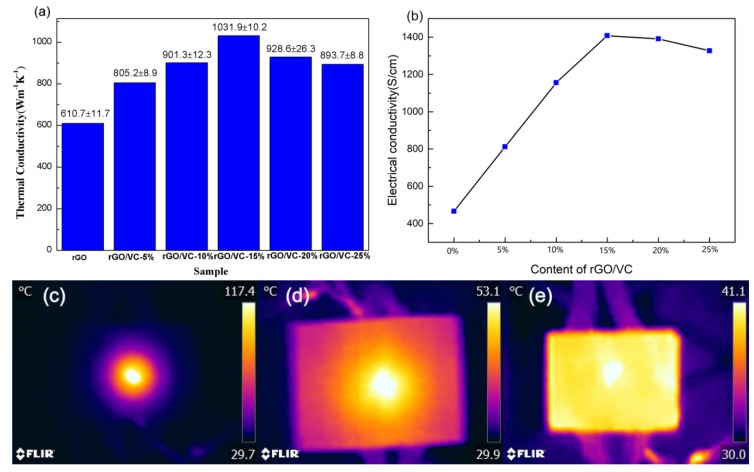
(**a**) In-plane thermal conductivity of rGO/VC-*x*% films; (**b**) electrical conductivities of rGO films with different contents of vitamin C; infrared surface thermography of (**c**) GO/VC-15%, (**d**) rGO and (**e**) rGO/VC-15% films.

**Figure 5 nanomaterials-10-00531-f005:**
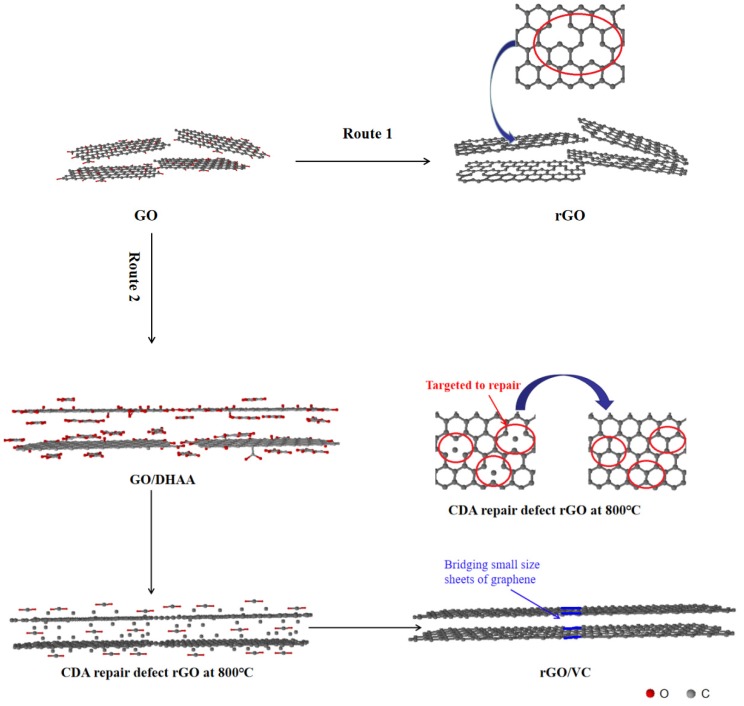
Mechanism of carbonized dehydroascorbic acid (CDA) for targeted repair of graphene defects and bridge connection of graphene sheets with small size.

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
