# Peer review of "Carbonized Dehydroascorbic Acid: Aim for Targeted Repair of Graphene Defects and Bridge Connection of Graphene Sheets with Small Size"

_nanomaterials, 2020, doi:10.3390/nano10030531_

Round 1
Reviewer 1 Report
In the present manuscript, the authors reported the optimisation of the GO reduction process by means the use of vitamin C that has been already reported in the literature for the same approach. the aim of the present work is the preparation of rGO layers with superior thermal conductivity for svariate applications. The manuscript is written taking care of details in the experimental procedures and discussion of the several characterization techniques reported by the authors. The topic is interesting and might attract the readers of the present journal. I recommend publication after revision and according to the comments below:
1) in the caption of figure 1b, please report the solvent utilised for the GO and rGO dispersions. Moreover, certain experimental detail were omitted being part of literature reported. I suggest to report details such as concentration of the reagents, solvents, temperature, etc...;
2) please add references where the authors discuss for the first time the utilisation of VC in the reduction process of GO, also reporting positive and negative issues with respect to the traditional more powerful methods (hydrazine for example);
3) in the discussion of the experimental results, more specifically Raman spectroscopy, the authors did not discuss if the evolution of CO2 due the thermal treatment had a role in the Id/Ig ratio. Please add comments.
4) how were the temperatures and time reported "Procedure temperature reduction rGO/VC films" selected? Is there any influence of the GO layer thickness on the reduction effectiveness?
5) the author should also report how electrical conductivities of the rGO layers changed before and after reduction. This data should flank (and agrees as well) with that of thermal conductivity.
Author Response
Dear reviewers, please see the attachment. We have studied comments carefully and have made correction accordingly.

Reviewer 2 Report
The manuscript by Li et al. introduces a novel method for the repair of defects in graphene sheets. This is an interesting method for production of highly conductive defect free graphene sheets. Compared to previous methods in the literature, the process of repair can be done under milder conditions and at lower annealing temperatures. The paper is well written and the conclusions are supported by the results. I recommend publication of this paper. I have only a short comment, to be addressed to before publication.
In the introduction, the authors have pointed out to the increase in the thermal conductivity of graphene/polyimide films due to the repair of defects in graphene. In such polymer films, the main factor affecting the thermal conductivity is the large Kapitza resistance at the interface (see Polymers 2019, ,11, 1465; doi:10.3390/polym11091465). This means that perhaps repairing the defects (to increase the thermal conductivity of graphene) does not improve the thermal conductivity of the film.
Author Response
Firstly, we are very grateful to you for recommending publication of this paper. On behalf of our co-authors, we would like to express our great appreciation to you. Considering your comment, our reply is as following:
As the reviewer has said that the main factor affecting the thermal conductivity is the large Kapitza resistance at the interface in such polymer films. However, a large number of phonon scattering is induced by structural defects in graphene membranes, which are mainly affected by grain size and interface resistence (Tailoring the Thermal and Mechanical Properties of Graphene Film by Structural Engineering. Small. 2018, 14, 1801346; doi: 10.1002/smll.201801346). The phonon interface scattering caused by the Kapitza resistance between graphene sheets greatly restricts the heat transfer performance of graphene films. Therefore, one of our research objectives was to reduce kapitza resistence by thermal decomposition of dehydroascorbic acid to fill the gaps between graphene sheets and hence improve the heat transfer performance of graphene films. On the other hand, according to relevant studies, it was found that the number and properties of defects on graphene sheets would affect grain size, thus greatly increase phonon scattering and reduce thermal conductivity (Structural defects in graphene. ACS Nano. 2011, 5, 26-41. doi: 10.1021/nn102598m; Thermal transport in graphene with defect and doping: Phonon modes analysis. Carbon, 2017, 116, 139-144. doi: 10.1016/j.carbon.2017.01.089). Reducing the number of defects on the graphene nanosheet can significantly improve the thermal conductivity of graphene (Strategy and mechanism for controlling the direction of defect evolution in graphene: preparation of high quality defect healed and hierarchically porous graphene. Nanoscale. 2014, 6, 13518-13526. doi: 10.1039/c4nr04453c; Healing of reduced graphene oxide with methane + hydrogen plasma. Carbon. 2017, 120, 274-280. doi: 10.1016/j.carbon.2017.05.032). Therefore, we further proposed a new strategy of utilizing dehydroascorbic acid for targeted defect repair to better restore the integrity of graphene sheets and improve the thermal conductivity of graphene membranes.
In summary, we aim to improve the thermal conductivity of the graphene membrane from the aspects of reducing Kapitza resistance and repairing defects. Finally deep thanks for your comments.
Round 2
Reviewer 1 Report
The authors have provided a convincing reply to comments raised in the original submission. The revised manuscript is now ready for publication.
This manuscript is a resubmission of an earlier submission. The following is a list of the peer review reports and author responses from that submission.
Round 1
Reviewer 1 Report
Nanomaterials
The article is about the targeted repair of graphene defects using Vitamin C.
The prepared rGO/VC was characterized using various techniques. The paper is well written and I recommend for publication.
Author Response
Point 1: More explanation is need for why the conductivity increase to maximum value with increase in VC content and then decreased? 

Response 1: Considering the Reviewer’s suggestion, we have made corrections according to the Reviewer’s comments. It can be observed that the conductivity rises up to maximum value with increment in VC content and then falls down, this is caused by the following reasons: As Vitamin C chemically reduces GO preliminarily they produces dehydroascorbic acid (DHAA). If vitamin C is not excessive, the more vitamin C is added, the more DHAA is produced. Carbonized dehydroascorbic acid nanoparticles (CDAN) from DHAA during thermal reduction are used to repair graphene defects and bridge graphene sheets. On the one hand, DHAA is decomposed thermally prior to GO to produce gases such as CO2, CO and H2O, which evenly spread out the GO sheet to form a gas passage to facilitate the emission of the GO thermal decomposition gas, thus reducing the chaotic stacking of the graphene sheets. However, the excess DHAA will produce extra atmospheric pressure, resulting in the formation of air holes between graphene sheets, which will destroy the order of graphene sheets. On the other hand, CDAN (generated during thermal decomposition of DHAA) not only repairs the vacancy defects of graphene but also produces amorphous carbon. The presence of an appropriate amount of amorphous carbon is conducive to bridging graphene sheets, enhancing interlaminar continuity and structural integrity, while excess amorphous carbon will hinder the repair of defects by CDAN and bridging graphene sheets. The well-arranged layered structure with fewer defects can minimize phonon scattering and thus improve the thermal conductivity of graphene films. Therefore, the conductivity first rises up to maximum value with increase in VC content and then falls down. (See page 10, lines 358-366.)
Reviewer 2 Report
In the current manuscript, Li et al. reported the targeted repair of graphene defects by using Vitamin C. Although the reduction of graphene oxide by vitamin C has already been reported, the authors are able to extent that work to repair the graphene defects, which can be applicable to solve the thermal dissipation issue of electronics devices. This approach is quite new and noteworthy to investigate. Therefore, I recommend acceptance of the manuscript after minor revision. My critical comments are given below –
The authors should correct the equations in Page No. 4. The authors should correct the English language. In the XRD pattern of rGO, the characteristic peak near the 2θ = 25Ëš is absent or less intense. The author should explain this anomalous behaviour. The authors should citesome recent articles on graphene like Nano Research 12 (11), 2655-2694, Progress in Energy and Combustion Science 75, 100786 etc.
Author Response
Point 1: The authors should correct the equations in Page No. 4. The authors should correct the English language. In the XRD pattern of rGO, the characteristic peak near the 2θ = 25Ëš is absent or less intense. The author should explain this anomalous behaviour. The authors should citesome recent articles on graphene like Nano Research 12 (11), 2655-2694, Progress in Energy and Combustion Science 75, 100786 etc. 

Response 1: We have made corrections according to the reviewer’s comments. The equations in Page No. 4 (line 181-187) have been corrected and the text has also been revised. In the XRD pattern of rGO, the characteristic peak near the 2θ=25Ëš is absent or less intense. This is because the mild thermal reduction temperature of 800℃ was adopted in this experiment, which was much lower than the thermal annealing temperature (above 2800 ℃) of traditional high thermal conductivity graphene membrane preparation. Although the thermal reduction temperature at 800℃ has an obvious cost advantage, GO is only carbonized and reduced, so the resulting rGO films does not have high crystallinity, and the diffraction peaks near 2θ=25°are not obvious, and the peaks are short and wide. In addition, since most of the oxygen-containing functional groups on GO are eliminated after thermal reduction treatment at 800℃, the blocking effect of oxygen-containing functional groups on the stacking of rGO sheets is reduced, so the graphene sheet spacing is reduced. According to the Bragg lattice equation, rGO diffraction peaks appear at 2θ=25.67°as the spacing between the crystal planes diminishes and theta enlarges (See page 9, lines 307-312). We carefully studied the two articles of Nano Research 12 (11), 2655-2694, and Progress in Energy and Combustion Science 75, 100786 recommended by reviewer 1, and got many useful inspirations, and cited the above two articles, which could be seen in [51] and [52] (See page 9, lines 307-312.).
Reviewer 3 Report
see file attached

Round 2
Reviewer 3 Report
Please see attached file
